# WHITENED SELF-ATTENTION

## ABSTRACT

Self-attention in Transformer architectures is formulated as a function of the pairwise contributions between target vectors and their context vectors. This construction implicitly assumes ternary and higher order relationships are negligible. It further treats the context vectors as though they can be processed individually, as if mutually independent of one another. This model contradicts, however, our understanding of language: that the meaning of words is influenced by complex interdependencies. We introduce *Whitened Self-Attention*, a theoretically motivated and novel enhancement that optimally accounts for inter-token correlations, and based on several covariance modeling assumptions, we derive a computationally feasible implementation for it. Experiments with a small GPT architecture show that whitened self-attention reduces perplexity by 19.3%, achieves the same mean cross-entropy loss in 37 times fewer training iterations, and, after hyperparameter optimization, reduces training time by 91%. Our approach shows significant potential for scaling and for improving the performance and generalization of large-scale language models. Moreover, as whitening decorrelates input sequences, it will affect the structure of the resulting trained attention and feedforward weight matrices. This will have an effect on their singular value decompositions, and should, in turn, influence the results of the many studies on the mechanistic interpretability of Transformers.

## 1   INTRODUCTION

The Transformer model Vaswani et al. (2017) is a popular and successful deep learning architecture used in a wide array of applications areas such as NLP Kalyan et al. (2021), computer vision Khan et al. (2022); Han et al. (2022), speech recognition Gulati et al. (2020), and computational biology Zhang et al. (2023). That said, the core component, self-attention, is more of a heuristic than a precisely formulated, optimally derived filter. The attention expression is usually computed in terms of query, key, and value vectors, but in this paper we find it advantageous to formulate it in terms of a statistical representation of target vectors, $x_n \in \mathcal{R}^d$, based on sets of weighted sums of their context vectors, $x_i \in \mathcal{R}^d$ Bahdanau et al. (2015). The causal autoregressive formulation used in GPT architectures takes the form

$$\text{Att}(x_n) = \sum_{i=0}^{n-1} \text{softmax}(\frac{x_n^\mathsf{T} Q^\mathsf{T} K x_i}{\sqrt{d}}) V x_i, \tag{1}$$

for $n = 1, \dots, N$, where $N$ is the size of the context window, and $Q$, $K$, and $V$ are learned matrices. The softmax terms in Equation 1 are positive scalars summing to one, and they function as intuitive, although ad-hoc, estimates of the relative information each context vector, $x_i$, contains about the target vector, $x_n$. As discussed in more depth in the following section, the $x_i$ would need to be independent random vectors for this formulation to provide a minimum variance estimator Shaffer (1991). If correlated the representation contains duplicated information and is necessarily suboptimal. When training with very large datasets, it is conceivable that the estimator's variance could approach an optimal value for frequently occurring tokens, but is unlikely to do so uniformly across the entire input vocabulary, especially rarer ones in the long tail. Whitening is a filtering process that transforms input sequences into stochastically decorrelated, normalized outputs, and methods based on it produce optimal, minimum variance, linear estimators Kleiner et al. (1979); Kailath (1970). The rest of this paper develops a computationally feasible whitening operator for self-attention, and presents experimental results demonstrating that when used to train a GPT model, whitened self-attention significantly improves performance and computation time.

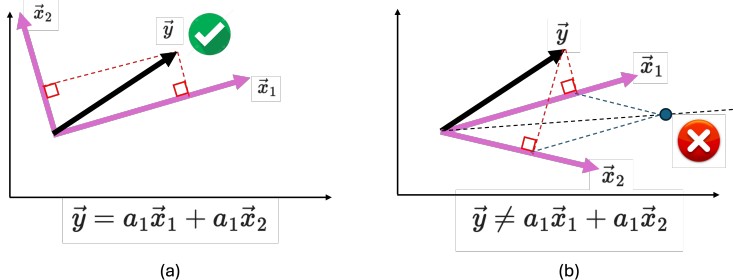

Figure 1: Part (a) shows three vectors, $\vec{y}$, $\vec{x_1}$, and $\vec{x_2}$ in $\mathcal{R}^2$, and illustrates that when $\vec{x_1}$ and $\vec{x_2}$ are orthogonal, $\vec{y}$ can be perfectly represented by a sum of projections. Part (b) shows that when $\vec{x_1}$ and $\vec{x_2}$ are not orthogonal, the sum of projections is the biased representation of $\vec{y}$ shown as the blue dot.

## 2  WHY WHITEN?

The self-attention mechanism is a statistical method for representing target vectors as weighted functions of their context vectors. As can be seen in the formulation provided by Equation 1, this is a pairwise operation, meaning the representational contributions to the vector, $x_n$, from any two vectors, $x_i$ and $x_j$, are made without taking into account their mutual interdependencies. Figure 1 provides a simplified explanation for why this matters. Part (a) of the figure shows that a vector $\vec{y}$ can be perfectly decomposed into a simple sum of its projections onto two orthogonal vectors $\vec{x_1}$ and $\vec{x_2}$. However, as shown in part (b) of the figure, when the two vectors are not orthogonal, the sum of the projections of $\vec{y}$ produces the highly biased result obtained by applying the vector parallelogram law, which in this case is the blue dot shown in the figure. Because the two vectors $\vec{x_1}$ and $\vec{x_2}$ are not orthogonal, they share components, and so the simple sum of the projections of $\vec{y}$ introduces a double-counting of information. For deterministic vector space problems, this duplication of information is eliminated using the Gram-Schmidt orthogonalization procedure Strang (2022), a method in linear algebra for taking a set of linearly independent vectors and constructing an orthonormal basis from them. That said, it is possible to achieve a learned orthogonalization result by introducing it as a criterion to minimize in a regularization term Xiao et al. (2024). For stochastic processes, a similar idea can be applied, but based on the orthogonalization of the covariance structure of the embedding vectors Papoulis & Pillai (2002); Haykin (2002). A perfect whitening transform makes the pairwise cross-covariance zero, linearly decorrelating them.

## 3  DERIVATION OF SEQUENCE WHITENING

In this section, we derive the whitening filter for the problem formulation corresponding to self-attention for a GPT model. Given an ordered sequence of column vectors $\{x_0, x_1, \ldots, x_{N-1}\}$, with $x_i \in \mathcal{R}^D$, the objective is to represent the next vector, $x_N$, given observations of the preceding context. This is a causal autoregressive estimation problem Akaike (1969). Typically, the $x_i$ are assumed (or modeled) as independent and identically distributed, but if the sequence is correlated it must be whitened to obtain the optimal linear estimator. Defining the column vector $X = [x_0^\intercal, x_1^\intercal, \ldots, x_{N-1}^\intercal]^\intercal \in \mathcal{R}^{ND}$, where the $x_i$ are zero-mean random vectors, the covariance matrix $\Lambda_X = E\{XX^\intercal\}$ has a symmetric block structure

$$\Lambda_X = \begin{bmatrix} \Lambda_{0,0} & \Lambda_{0,1} & \ldots & \Lambda_{0,N-1} \\ \Lambda_{1,0} & \Lambda_{1,1} & \ldots & \Lambda_{1,N-2} \\ \vdots & \vdots & & \vdots \\ \Lambda_{N-1,0} & \Lambda_{N-1,1} & \ldots & \Lambda_{N-1,N-1} \end{bmatrix}, \tag{2}$$

where $\Lambda_{i,j} = E\{x_i x_j^\intercal\}$. The whitened sequence, $W = [w_0^\intercal, w_1^\intercal, \ldots, w_{N-1}^\intercal]^\intercal \in \mathcal{R}^{ND}$, is obtained from $X$ with the transformation $W = \Lambda_X^{-1/2} X$. That the $w_i$ are white, meaning their cross covariance matrix is the identity matrix, can be verified as follows:

$$\Lambda_W = E\{WW^\intercal\} = E\{\Lambda_X^{-1/2} XX^\intercal \Lambda_X^{-1/2}\} = \Lambda_X^{-1/2} \Lambda_X \Lambda_X^{-1/2} = I. \tag{3}$$

The set of whitened vectors, $\{w_i\}$, span the same subspace as the $\{x_i\}$ but in the mean are normalized and, assuming they are jointly Gaussian, are independent of each other. When substituted into the self-attention expression in Equation 1, the result is what we call whitened self-attention (WSA),

$$\text{WSA}(x_n) = \sum_{i=0}^{n-1} \text{softmax}\left(\frac{x_n^\intercal Q^\intercal K w_i}{\sqrt{d}}\right) w_i. \tag{4}$$

An important difference between this expression and the one for standard attention in Equation 1 is that there is no $V$ matrix. As will be explained in more detail in the next section, it has been absorbed into the $w_i$.

## 4 MODELING THE COVARIANCE STRUCTURE

As the matrix $\Lambda_X$ is $ND \times ND$, its inverse is computationally challenging and memory intensive. For example, in current production-quality LLMs, $ND \approx 10^7$, meaning this single matrix could require roughly a petabyte of memory. Some of the computational and memory requirements can be mitigated by modeling the structure of $\Lambda_X$. It is common to model the sequences as being wide-sense stationary, which makes the cross-covariance blocks, $\Lambda_{ij}$, dependent only on their indicial separation, $|i - j|$ Van Trees (2004); Papoulis & Pillai (2002). Application of the condition makes the structure of $\Lambda_X$ block Toeplitz:

$$\Lambda_X = \begin{bmatrix} \Lambda_0 & \Lambda_1 & \dots & \Lambda_{N-1} \\ \Lambda_1 & \Lambda_0 & \dots & \Lambda_{N-2} \\ \vdots & \vdots & & \vdots \\ \Lambda_{N-1} & \Lambda_{N-2} & \dots & \Lambda_0 \end{bmatrix}. \tag{5}$$

The structure can be further simplified if we assume the covariance process can be modeled as having compact support, meaning that the covariance is zero for values of $|i - j|$ greater than a specified value. For example, if $\Lambda_k = 0$ for $k = |i - j| > 0$ then $\Lambda_X$ is block diagonal, and the whitening filter, $\Lambda_X^{-1/2}$, is also block diagonal. This makes the whitened vectors $w_i = \Lambda_0^{-1/2} x_i$, and we can associate the $V$ matrix in Equation 1 as being $\Lambda_0^{-1/2}$. This observation clarifies why the expression in Equation 4 need not explicitly represent $V$, because, by construction, it is contained in the $w_i$. Moreover, the association of $V$ to $\Lambda_0^{-1/2}$ provides the useful interpretation that $V$ is a zero$^{\text{th}}$-order whitening filter, acting to decorrelate the internal components of the embedding vectors. The less trivial case is when $\Lambda_k = 0$ for $k > 1$, making $\Lambda_X$ block tridiagonal,

$$\Lambda_X = \begin{bmatrix} \Lambda_0 & \Lambda_1 & & & \\ \Lambda_1 & \Lambda_0 & \Lambda_1 & & \\ & \Lambda_1 & \Lambda_0 & \Lambda_1 & \\ & & \ddots & \ddots & \ddots & \\ & & & \Lambda_1 & \Lambda_0 & \Lambda_1 \\ & & & & \Lambda_1 & \Lambda_0 \end{bmatrix}. \tag{6}$$

As $\Lambda_X$ is symmetric and positive semi-definite, it can be factored with a block Cholesky decomposition, $\Lambda_X = LL^\intercal$, where L is a block lower triangular matrix Golub & Van Loan (2013) and $^\intercal$ is the transpose operator. In this case, $L$ is block bidiagonal:

$$L = \begin{bmatrix} L_1 & & & & \\ M_2 & L_2 & & & \\ & M_3 & L_3 & & \\ & & \ddots & \ddots & \\ & & & M_{N-1} & L_{N-1} \\ & & & & M_N & L_N \end{bmatrix}. \tag{7}$$

Replacing $\Lambda_X^{-1/2}$ with $L^{-1}$ in Equation 3 confirms that $W = L^{-1}X$ is white. This expression consists of $2N - 1$ blocks, but recognizing that the matrix represents a recursion, we assume the sequence of $L_i$ and $M_i$ will converge to steady state values, $L_\infty$ and $M_\infty$ for sufficiently large $i$,

Brockwell & Davis (1991). The structure of $L$ can then be simplified with the approximation that substitutes all the matrices in Equation 7 with their steady-state terms, making it

$$L = \begin{bmatrix} L_\infty & & & & & \\ M_\infty & L_\infty & & & & \\ & M_\infty & L_\infty & & & \\ & & \ddots & \ddots & & \\ & & & M_\infty & L_\infty & \\ & & & & M_\infty & L_\infty \end{bmatrix}. \tag{8}$$

With this form, it is a straight-forward application of block Gaussian elimination to solve for $W = L^{-1}X$, which yields the following simple recursion for the whitening filter:

$$w_0 = L_\infty^{-1}x_0, \quad w_i = L_\infty^{-1}(x_i - M_\infty w_{i-1}), \quad i = 1, \dots, N-1. \tag{9}$$

This process is represented graphically in the lower part of Figure 2, which illustrates how each succeeding vector input in the sequence is whitened as a transformation of the linear combination with the previous vector output of the recursion. The elements of the matrices $L_\infty^{-1}$ and $M_\infty$ are learned directly as part of training the Transformer model. $L_\infty$ itself need not be known or inverted. These covariance modeling steps represent a series of engineering approximations that allow us to construct a practical implementation of whitened self-attention. In the following section, we present experimental results illustrating the performance of this whitened self-attention formulation.

## 5 EXPERIMENTS

To test the effect of our whitened self-attention procedure, we implemented it using Equations 4 and 9 and compared its results with those from standard attention, per Equation 1. To focus our study primarily on the effects of the whitening filter, we opted to limit the number of layer interactions to the minimum needed. Our experimental design was based on the small Transformer architecture shown in Figure 2.[1] It is a GPT model consisting of two Transformer decoder blocks, each containing a two-head attention layer and a standard 4x-fanout feedforward layer. Positional information was incorporated using RoPE Su et al. (2024). The remaining elements: layer norms, linear projections, embedding, and unembedding layers are standard components.[2]

The data used in the experiments is the collected works of Charles Dickens, obtained from the Project Gutenberg website Dickens (2018). The data were used without any additional preprocessing. The corpus contains 13.8m characters, and we opted to use a character-based tokenization strategy Banar et al. (2020). The advantage of this choice is that it explicitly limited the vocabulary size to the number of unique characters in the data, which for this case is 93. Tokenization did not require ad-hoc mapping of infrequent tokens to a special placeholder, and it precluded the need to experiment with a hyperparameter optimization to find the appropriate vocabulary size, as would have been required using a more sophisticated tokenization scheme such as BPE Gage (1994); Sennrich et al. (2016) or WordPiece Schuster & Nakajima (2012). Our experimental setup specified a token embedding dimension of 256 and a context window of length 256, which was chosen to be roughly three times the average paragraph length in our corpus. As our comparative metric of performance, we used the mean cross-entropy (MCE) loss applied to the validation data. The experiments used 90% of the Dickens data for training with 10% set aside for validation.

With these details, we ran four experiments, each for 100k iterations with a batch size of 256,[3] corresponding to roughly three epochs. Our baseline was the standard GPT implementation of self-attention with RoPE, run without whitening. Given the hyperparameters described in the preceding discussion, the model size worked out to 1.62m weights, and the MCE loss on the validation data during optimization is shown by the blue curve in Figure 3. The final value of MCE loss for this experiment was 1.39, corresponding to a perplexity of 4.0 (versus 93 for the untrained model). These results are summarized in the first row of Table 1, which also provides the experiment's total compute time of 57 minutes.[4] The green curve in Figure 3 is the result for the same experiment, using the

---

[1] Complete code and data will be released to GitHub upon publication

[2] See https://transformer-circuits.pub/2021/framework/.

[3] Except for the last experiment, which used a batch size of 128.

[4] All computations were performed on an Nvidia RTX 4090

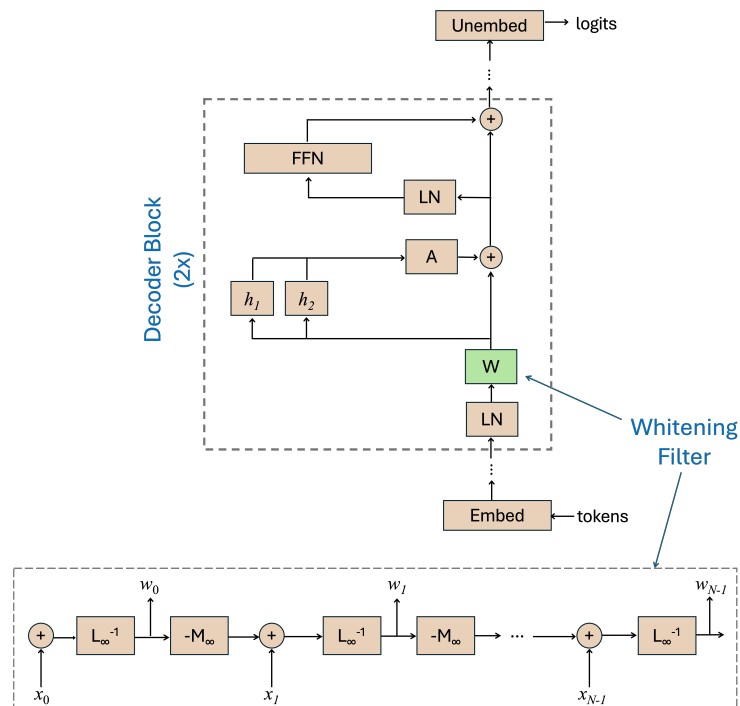

Figure 2: Decoder Transformer architecture with two blocks. Each block consists of two attention heads ($h$), one feedforward layer (FFN), two layer norms (LN), and a projection (A) that combines the outputs from the attention heads. The whitening filter, W, implements the recursion in Equation 9. The whitened sequence passes through the attention layer, and is also passed forward through the residual stream. The embedding layer converts the input tokens to vectors, and the unembedding layer outputs the logits.

same hyperparameters, but with the addition of the whitened self-attention recursion block, shown in Figure 2. Due to the two matrices, $L_\infty^{-1}$ and $M_\infty$, one pair used with each decoder block, the model size increased by four times the squared embedding dimension (256) to 1.88m parameters. As can be seen in the Figure 3, the performance of this model rapidly outstrips standard attention, dropping to the same MCE loss in just 2,103 iterations (a factor of $48\times$ less), and requiring only 9.6 minutes of compute time (83% faster). The loss value continues to drop, achieving a value of 1.16 at 100k iterations. This corresponds to a perplexity of 3.18, a 21% improvement in performance over the result for standard attention. These results are recapped in the second row of Table 1, which also shows the total compute time was 457 minutes.

As the whitened self-attention experiment benefited from more parameters, we ran a third experiment for standard attention. This experiment used an equivalent model capacity of 1.88 million parameters, achieved by increasing the token embedding dimension from 256 to 276. All other hyperparameters, including the batch size, were held the same. The results are represented by the orange curve in Figure 3. The final MCE loss for this trial was 1.38, an insignificant improvement over the smaller, baseline, standard attention experiment. Due to the increased model size, the experiment required more time, completing in 71 minutes. These results are shown in the third row of Table 1. By comparison, the whitened self-attention experiment achieved the same level of loss in 2,362 iterations (42 times fewer, corresponding to 86% faster), while delivering a 19.7% improvement in perplexity at 100k iterations.

It can be seen from the second and third rows of Table 1 that the whitened self-attention model required 6.4 times more compute time per iteration than the standard attention model with an equivalent model size, but as discussed, it achieves the same level of MCE loss in many fewer iterations and much less compute time. This motivated a fourth experiment, rerunning the whitened self-attention model with half the batch size. This modification has no effect on the model size, but allows the training loop to achieve the 100k iterations in roughly half the time (and roughly half the number of epochs). The

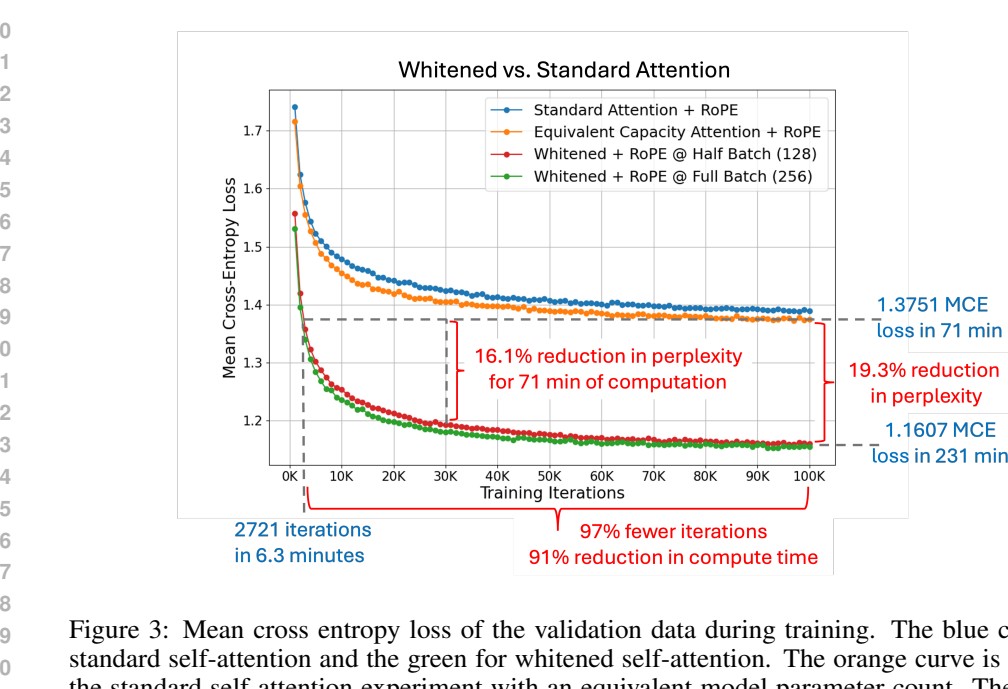

Figure 3: Mean cross entropy loss of the validation data during training. The blue curve is for standard self-attention and the green for whitened self-attention. The orange curve is a repeat of the standard self-attention experiment with an equivalent model parameter count. The red curve is a repeat of whitened self-attention using half the batch size. Annotations compare whitened self-attention computed with half batch to the standard self-attention with equivalent model capacity.

results are represented by the red curve in the Figure 3. The final MCE loss is almost identical to the whitened experiment with the larger batch size, however, as shown in the fourth row of Table 1, it completed in 51% of the compute time. In comparison with the equivalent capacity standard attention experiment, it achieves the same level of MCE loss at iteration 2,721, corresponding to a 91% reduction in compute time.

Table 2 recaps the compute time comparisons of the equivalent capacity standard attention model with the full-batch and half-batch whitened self-attention experiments. The key comparisons of the equivalent capacity standard self-attention with the half-batch whitened self-attention are annotated on the curves in Figure 3. Although obtained with a small corpus and a small model, these experiments demonstrate the potential whitening has to improve the representational power of self-attention, or alternatively to reduce the compute time for an equivalent MCE loss. The annotations on Figure 3 further show that whitening provides a spectrum of engineering options ranging between an equivalent performance in less time to improved performance in an equivalent time. Thus, for the equivalent compute time of 71 minutes, the half-batch whitened model delivers a 16.1% improvement in perplexity over the standard attention with equivalent capacity.

| Experiment | Model Size | MCE Loss | Perplexity | Compute Time |
|---|---|---|---|---|
| Standard Attention | 1.62m | 1.39 | 4.00 | 57 min |
| Whitened Attention | 1.88m | 1.16 | 3.18 | 457 min |
| Equivalent Capacity | 1.88m | 1.38 | 3.96 | 71 min |
| Whitened Half Batch | 1.88m | 1.16 | 3.19 | 231 min |

Table 1: Performance summary of experiments comparing standard vs. whitened self-attention at 100k iterations. Equivalent Capacity in row three refers to the standard attention experiment with more parameters and the Whitened Half Batch in row four refers to the whitened experiment run with a smaller batch size.

We wanted to gauge how whitened self-attention scales with changes to the sequence length and embedding dimension, and this is shown in Table 3a. In the two leftmost columns, the table show the

| Model | Batch Size | Iters | Compute Time | %Time |
|-------|------------|-------|--------------|-------|
| ECSA | 256 | 100k | 71 min | 100% |
| WSA | 256 | 2,103 | 9.6 min | 17% |
| WSAHB | 128 | 2,721 | 6.2 min | 9% |

Table 2: Recap of the number of iterations and compute time needed to attain an MCE loss of 1.38: ECSA is for Equivalent Capacity Standard Attention and is the baseline, WSA is for Whitened Self-Attention, and WSAHB is for Whitened Self-Attention Half Batch. Each model was trained with 1.88m weights.

sequence length, and embedding dimension used, and in the two rightmost columns, the resulting model size in millions of weights and the compute time in milliseconds per training iteration. For comparison, Table 3b provides the same information for standard attention. The tables show that for both whitened and standard attention, it is the embedding dimension that most affects the model size. In both cases, doubling the embedding dimension roughly quadruples the model size, as expected due to its the quadratic relationship to the attention matrices. Although the compute time for whitened self-attention is roughly 10 times that of the standard model, both change linearly, in that doubling either the sequence length of the embedding dimension roughly doubles the amount of compute time.

| Sequence Length | Embedding Dimension | Model Size | Compute Time | | Sequence Length | Embedding Dimension | Model Size | Compute Time |
|-----------------|---------------------|------------|--------------|---|-----------------|---------------------|------------|--------------|
| 256 | 256 | 1.88m | 175ms | | 256 | 256 | 1.62m | 16ms |
| 512 | 256 | 1.88m | 324ms | | 512 | 256 | 1.62m | 37ms |
| 256 | 512 | 7.44m | 352ms | | 256 | 512 | 6.39m | 36ms |

(a)  (b)

Table 3: Scaling comparison of (a) whitened attention and (b) standard self-attention across different sequence lengths and embedding dimensions. Model size is shown in millions of weights, and compute time in milliseconds per training iteration.

## 6 EVALUATION OF THE WHITENING PROCESS

The results of Section 5 show that our whitening process effectively accelerates learning, which, in turn, implies it is succeeding in reducing the duplication of information, but it remains to show that the filter is effectively whitening the input sequences. To study this question, we computed the sample covariances of the inputs and outputs of the whitening filter. Effective whitening would mean that the elements of the output sample covariance would concentrate a higher percentage of energy on the main diagonal than does the input matrix. Given a batch of input sequences in the form of a tensor $x$ with shape $B \times T \times D$, where $B$ is the batch dimension, $T$ the sequence length, and $D$ the embedding dimension, $x_{bt}$ for $b \in \{1, \ldots, B\}$ and $t \in \{1, \ldots, T\}$ is the $D$-dimensional vector at sequence position $t$ in batch item $b$. The sample covariance matrix, $\tilde{\Lambda}_X$, can be computed over the batch dimension from the stacked vectors, $X_b = [x_{b1}^\intercal, \ldots, x_{bT}^\intercal]^\intercal \in \mathcal{R}^{TD}$:

$$\tilde{\Lambda}_X = \frac{1}{(B-1)} \sum_{b=1}^{B} (X_b - \tilde{\mu}_X)(X_b - \tilde{\mu}_X)^\intercal, \tag{10}$$

where $\tilde{\mu}_X$ is the corresponding sample mean. Using similar notation, the sample covariance for the whitened sequences, $W_b = [w_{b1}^\intercal, \ldots, w_{bT}^\intercal] \in \mathcal{R}^{TD}$, is

$$\tilde{\Lambda}_W = \frac{1}{(B-1)} \sum_{b=1}^{B} (W_b - \tilde{\mu}_W)(W_b - \tilde{\mu}_W)^\intercal. \tag{11}$$

The covariance matrix of a perfectly whitened sequence has ones on the diagonal and zero elsewhere, so we can define a measure of whiteness as the ratio, $\Psi_W$, of the sum of magnitudes of elements not on the diagonal to those that are.

$$\Psi_W = \frac{1}{(TD-1)} \sum_{i \neq j} |\tilde{\Lambda}_{W_{ij}}| / \sum_i |\tilde{\Lambda}_{W_{ii}}|, \tag{12}$$

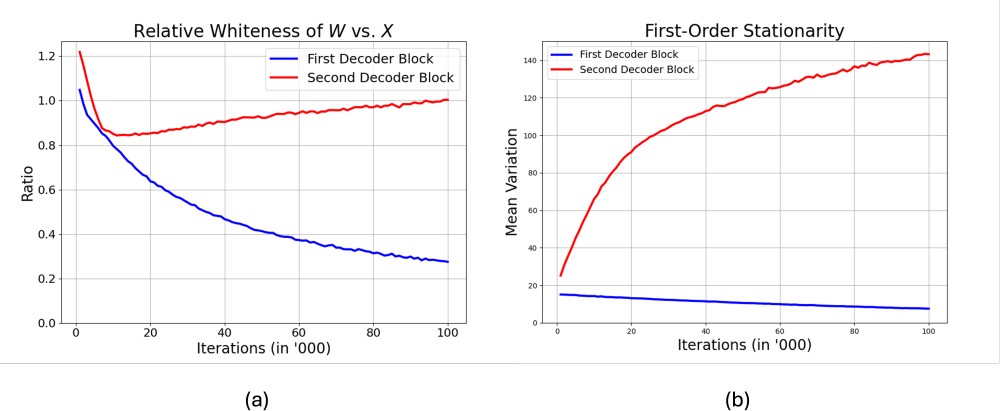

(a)                                                                          (b)

Figure 4: (a) Relative whiteness of $W$ with respect to $X$ as a function of training iterations for the whitened self-attention experiment using full batch. (b) Measure of first-order stationarity as a function of training, showing the inputs to the first decoder block have begin with and improve a good degree of first-order stationarity, whereas the second becomes increasingly non-stationary.

where the summation indices $i$ and $j$ range over $1, \ldots, TD$, and the term $(TD - 1)$ accounts for the fact that there are only $TD$ terms used in the denominator versus the $TD(TD - 1)$ in the numerator. The ratio would be closer to zero for whiter covariances, however, due to the approximations made in modeling the covariance, we do not expect the filter to perfectly whiten the sequences, and so experimentally we evaluate the measure of the relative whitening of $W$ with respect to $X$. Figure 4a shows the computation of $\Psi_W / \Psi_X$ as a function of training iterations for the full-batch, whitened self-attention experiment described in the Section 5. The blue curve shows the relative whitening result for the first decoder block, and the red curve is for the second. For the first decoder block, it shows that the training process learns to significantly whiten its input, reducing the relative whitening ratio to a value of 0.29, an improvement of over 70%.

In contrast, after initially whitening its input to 83%, with additional iterations, the second decoder block reverses direction, trending back to unity. To explain why the first block whitens while the second does not, we study a key assumption used in deriving the whitening filter: that the input sequences can be reasonably modeled as first-order autoregressive processes (Equation 6). This can be evaluated experimentally by computing the first-order sample cross-covariance of the input sequences across the batch dimension. For $x_{b,t} \in \mathcal{R}^D$, the first-order sample cross-covariance can be computed as

$$\tilde{\Lambda}_{t,t+1} = \frac{1}{B-1} \sum_{b=1}^{B} (x_{b,t} - \tilde{\mu}_{x_t})(x_{b,t+1} - \tilde{\mu}_{x_{t+1}}), \tag{13}$$

where $\tilde{\mu}_{x_t}$ is the corresponding sample mean. Given these statistics, a measure of first-order stationarity can be computed as the variation of the $\tilde{\Lambda}_{t,t+1}$ around their mean for $t = 0, \ldots, T-1$:

$$\rho = \sum_{t=0}^{T-1} ||\tilde{\Lambda}_{t,t+1} - \mu||_F, \text{ where } \mu = \frac{1}{T} \sum_{t=0}^{T-1} \tilde{\Lambda}_{t,t+1}, \tag{14}$$

and $|| \cdot ||_F$ designates the Frobenius norm. Inputs that ideally correspond to a first-order stationary sequence would have $\rho = 0$. Figure 4b, illustrates the statistic in Equation 14, an estimate of the first-order stationarity of the input sequences to each whitening filter as a function of training iterations. The value of $\rho$ for input sequences to the first block begins at a value of 18, reducing to 10 at the end of training. The result for the second block, however begins at 24, increasing to over 140. This experimental evidence shows that the input sequences to the second whitening filter are not first-order stationary, explaining why the second block is unable to whiten its inputs, as illustrated by the red curve in Figure 4a.

## 7 DISCUSSION AND FUTURE RESEARCH

The trend in deep learning is to publish results based on large models and datasets, and although this provides some reassurance about the potential for generalization, it can also obscure the relationship between theory and experimental results. We have deliberately chosen to work with a small model and dataset, which has allowed us to design experiments with more control and more readily interpretable results. It has helped isolate and understand the effects of specific architectural and covariance modeling choices, and importantly, it should make it easier for others to reproduce, verify, and build upon our work. This paper demonstrates that whitening improves the performance of self attention within the context of our GPT architecture. At convergence, it delivers a 20% improvement in perplexity, and it achieves standard attention's best result in 91% less time even though it requires the implementation of a recursion. If these results carry over to larger LLM models it would mean considerable savings in compute time, improvements in performance, or some combination of both. We further show that although the loss function is applied to the output of the fully composed model, the first-order stationary components of the input sequences undergo proper whitening. Finally, our experiments show that the MCE loss for whitened self-attention converges in orders of magnitude fewer iterations than that for standard attention. This suggests that standard self-attention is fundamentally inefficient at learning and reinforces the theoretical notion that the representation of target vectors from their context is improved when duplicative information is removed through a process of covariance orthogonalization.

As seen from the experiments, training the half-batch whitened self-attention model required $3.25\times$ more time per iteration than equivalent capacity standard attention. This ratio was similar in inference, requiring 3.1x more time in a test to generate 10,000 tokens. Thus, a legitimate concern is how these results vary for much longer context windows and larger embedding dimensions. The inversion of the Cholesky factor is based on a recursion whose computational complexity naturally grows with the length of the sequence. Moreover, recursions are generally considered not well-suited for implementation on GPUs. That said, the parallelization primitive known as the method of prefix sums is, in principle, ideally suited to this type of problem. The whitening filter in Equation 9 can be reformulated as an associative matrix product, and Blelloch's algorithm Blelloch (1990); Harris et al. (2007); Kogge & Stone (2009) and the Hillis-Steele method Hillis & Steele Jr (1986) can be applied to inverting the Cholesky factor, computing a recursion of length $N$ in $O(\log N)$ steps, requiring $O(N)$ work using $O(N)$ processors. This has the potential to accelerate the efficiency of the whitened self-attention model for both training and inference, but may have trade-offs with the memory management of the matrix and batch dimensions.

The results of this paper sketch a clear roadmap for the next phase of our research. The first step will be to focus on scaling to larger corpora, using more sophisticated tokenization strategies, and implementing larger GPT models. Part of this task will include the implementation of Blelloch's prefix scan algorithm. In one of our experiments, we saw a simple change in batch dimension significantly reduced compute time, and hyperparameter optimizations are an additional task we foresee. Our mathematical developments highlighted the importance of covariance modeling, and we plan to explore additional options at both the global and block levels. For covariance matrices at the block level, matrix series truncations and approximations of various types have the potential for reducing memory and accelerating computation. These include methods such as Neumann expansions, Krylov subspaces Strang (2000), and diagonal plus low rank matrices Saunderson et al. (2012). At the global covariance matrix level, extending the block tridiagonal model in Equation 6 to higher orders, for example the pentadiagonal case, seems a promising direction of enquiry for improving the whitening process. Finally, a great many papers have used singular-value decompositions to study the characteristics of attention and feedforward matrices. Our results demonstrate that the relationships between input vectors are more complex than can be captured by standard attention alone, and it would not be surprising that SVDs of trained weight matrices based on unwhitened inputs would be biased by their correlations. This opens a spectrum of potential research directions, revisiting some of the key influential papers in this domain with whitened input sequences in place of the standard ones. Examples include the analysis low-rank structures Lan et al. (2020); Wang et al. (2020), studies on attention head specialization Voita et al. (2019); Brody et al. (2023), the disentanglement of multi-head attention contributions Michel et al. (2019), the implementation of test time pruning He & Lin (2025), the study of context-specific behaviors Yao et al. (2024), and the general subject of mechanistic interpretability Bereska & Gavves (2024); Frankle & Carbin (2018); Naim & Asher (2024); Scherlis et al. (2022).

ACKNOWLEDGMENTS

Removed for anonymity.

REPRODUCIBILITY STATEMENT

All of the results presented in this paper can be reproduced with the data and code provided in the supplementary materials, which will also be made publicly available in the first author's GitHub repository upon publication. Reported timings are based on an NVIDIA RTX 4090 GPU, and will likely vary when using alternative hardware.

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
