# OpenReview forum: "Whitened Self-Attention"
_ICLR.cc/2026/Conference — Submitted to ICLR 2026_

### Official Review · Reviewer_g2W4 · 2025-10-26

**Soundness:** 2
**Presentation:** 3
**Contribution:** 3
**Rating:** 4
**Confidence:** 4

**Summary:**

This paper proposed a modified self-attention mechanism motivated by whitening the input features of the self-attention. The authors claim that empirically the it improves over normal self-attention on a 2-layer GPT model.

**Strengths:**

The idea is straightforward and simple to understand. The motivation (whitening) is also less discussed for self-attention so this study could potentially add research diversity to the field.

**Weaknesses:**

For the technical correctness:
 - I am not sure why on line 161, you can assume the $L_{i}$ and $M_{I}$ converges to a steady state values, and also, why you can replace every recursion with your assumed steady state $L_{\infty}$, $M_{\infty}$. Please elaborate more on these, this doesn't look trivial to me.
 - Following the above point, after the simplification made with the assumption of the authors, I am not sure how is the model trained/encouraged to whiten the input sequence. For equation 9, without any regularization loss for learning the $L_{\infty}$, $M_{\infty}$ matrices, how do you encourage them to converge to the ground truth $L$ that whitens the input? The authors should elaborate more on this.

For the experiments:
 - I don't think Figure 4 well supports the authors claim about the functionality of the whitening filter. It is true that from (a), it shows decreased correlation as training progresses. However, it should be compared with a baseline without the proposed whitening filter. Since the token embedding and the layer norm are trained, it could be that they naturally learn to be de-correlated with training.
 - From Table 3, it seems like the author did run experiments on 7M models. Why not also post the results of MCE for the 7M model? Now I am not sure if the model scales well given the super small model size.
 - From the experiments (Figure 3), it is unclear to me if the model overfits or underfits without the training set MCE. Thus it is also unclear to me if the methods helps generalization, or it helps the model better fits the dataset. E.g., if the training MCE curves are the similar for both methods, then the proposed methods improves majorly generalization. The authors should clarify this.

**Questions:**

See the above weaknesses.

---

> ### Author Response · Authors · 2025-11-20
> **Clarifications on Steady-State Assumptions, Whitening Convergence, and Experimental Validation**
>
> ## Weaknesses:
>
> For the technical correctness:
>
> I am not sure why on line 161, you can assume the $L_i$ and $M_i$ converges
> to a steady state values, and also, why you can replace every
> recursion with your assumed steady state $L_{\infty}$, $M_{\infty}$. Please elaborate more on
> these, this doesn't look trivial to me.
>
> > We are not assuming, we are approximating.  It is a fact that ARMA processes generate a transient plus a stable
> steady-state behavior when the underlying transfer function of the
> corresponding linear time-invariant model has eigenvalues with
> magnitude less than one.  Using only the steady-state values is an
> approximation that ignores the transient, a common (but not required)
> engineering practice in control theory.  The benefit of this assumption is that it significantly reduces
> the number of additional parameters to learn.  In closed form, the solution
> to the steady state is found by solving the algebraic Riccati
> equation, but in the context of deep learning, we solve it as part of
> the gradient descent problem.  We don't expect everyone to be
> knowledgeable about this, but we cite a classic reference on the topic
> (Peter J Brockwell and Richard A Davis. Time Series: Theory and
> Methods. Springer science & business media, 1991).  Would an appendix
> covering these topics make this paper more acceptable?
>
> Following the above point, after the simplification made with the
> assumption of the authors, I am not sure how is the model
> trained/encouraged to whiten the input sequence. For equation 9,
> without any regularization loss for learning the Linf, Minf, matrices,
> how do you encourage them to converge to the ground truth that whitens
> the input? The authors should elaborate more on this.  For the
> experiments:
>
> I don't think Figure 4 well supports the authors claim about the
> functionality of the whitening filter. It is true that from (a), it
> shows decreased correlation as training progresses. However, it should
> be compared with a baseline without the proposed whitening
> filter. Since the token embedding and the layer norm are trained, it
> could be that they naturally learn to be de-correlated with training.
>
> > Originally we expected it would be necessary to construct an
> auxiliary loss function for the $L_{\infty}$ and $M_{\infty}$ matrices,
> but we discovered that they converge without it. Figures 4a and 4b are
> the proof of this.  Figure 4a
> compares the input/output relationship of the token sequence across the whitening filter.  It shows that the sequence becomes 75% whiter.  This in conjunction with the
> accelerated descent of the loss function for the model that incorporates whitening is compelling numerical proof, and we don't understand the rationale of your objection.
>
> From Table 3, it seems like the author did run experiments on 7M
> models. Why not also post the results of MCE for the 7M model? Now I
> am not sure if the model scales well given the super small model size.
>
> > The models in Table 3 were run for just a few thousand iterations,
> which enabled us to work out the time and memory scaling
> relationships.  That said, our focus in this paper is the development of a
> theoretical model.  We are not making claims about its ability to generalize to other
> models.
>
> From the experiments (Figure 3), it is unclear to me if the model
> overfits or underfits without the training set MCE. Thus it is also
> unclear to me if the methods helps generalization, or it helps the
> model better fits the dataset. E.g., if the training MCE curves are
> the similar for both methods, then the proposed methods improves
> majorly generalization. The authors should clarify this.
>
> > Fair enough.  Thanks for your feedback on this.

---

> > ### Comment · Reviewer_g2W4 · 2025-11-21
> >
> > I thank the author(s) for their response and clarification. While the response did help me better understand the work, some of my concerns still remain, which I will elaborate further.
> >
> > Based on my understanding, your design of whitening filter (Equation 9), is motivated by whitening process, but there is no additional regularization on the learnable weights to encourage whitening. Then for it to work, the assumption is that by minimizing the MCE loss, the gradient descend will naturally make the filter perform whitening, which the authors also empirically verified with Figure 4.
> >
> > Given this, my main concern is about the necessity of adopting Equation 9 over other simple neural layers if training with MCE loss naturally encourages whitening. For instance, the Equation 9 seems to resemble a RNN architecture. Given that there is no constraint on the filter weights, how does this design compare to simply adding a layer of RNN before standard attention?
> >
> > Ideally, you would like to show that Equation 9 will learn better whitening/have better performance over baselines such as simply use a one-layer LSTM/GRU, or even a simple Linear layer or CNN before the standard attention. This shows your filter design is indeed more effective than existing simple architectures. For now, it is not clear to me why these architectures couldn't learn to whiten while your filter design could, given that both are trained only with MCE.
> >
> > Additionally, for the first point, the use of the word "assume" comes from line 161 in the paper. I advise the authors to make revisions based on the response.

---

> > > ### Author Response · Authors · 2025-11-21
> > >
> > > ...my main concern is about the necessity of adopting Equation 9 over other simple neural layers if training with MCE loss naturally encourages whitening. For instance, the Equation 9 seems to resemble a RNN architecture. Given that there is no constraint on the filter weights, how does this design compare to simply adding a layer of RNN before standard attention?
> > >
> > > Ideally, you would like to show that Equation 9 will learn better whitening/have better performance over baselines such as simply use a one-layer LSTM/GRU, or even a simple Linear layer or CNN before the standard attention. This shows your filter design is indeed more effective than existing simple architectures. For now, it is not clear to me why these architectures couldn't learn to whiten while your filter design could, given that both are trained only with MCE.
> > >
> > >
> > > > Yes, I'd say your remark is insightful, and it is conceivable -- even likely that an RNN could deliver similar results.  The difference is that our theoretical development allows us to connect intuition with architectural implementation.  That should help guide us on tradeoffs between performance and cost when selecting the hyperparameters for this component.  For example, if the performance is not as good as desired, Equation 6 can be adjusted from a block tridiagonal to a block pentadiagonal matrix, corresponding to a more sophisticated ARMA model.  The recursion in Equation 9 then becomes second order.  It also provides the insights needed to test the results against the assumptions, as was done in Figure 4.
> > > The hyperparameters of an RNN can also be adjusted, but it would be trial and error and without the benefit of the intuition derived from the theoretical derivation, and without clarity on what could be tested to evaluate how it functions.
> > >
> > > > We believe that this type of theoretical modeling is important because one of the main issues with Transformer architectures is the lack of explainability.  Our work is a step in the explainability direction.
> > >
> > > Additionally, for the first point, the use of the word "assume" comes from line 161 in the paper. I advise the authors to make revisions based on the response.
> > >
> > > > That's a good point.  For us, the word "assume" implies "assume the following approximation holds" or "assume the following model holds", but thanks to your feedback, we see that not all readers will understand that implicit meaning, and we will edit the text as you suggest.

---

### Official Review · Reviewer_3wfN · 2025-10-29

**Soundness:** 3
**Presentation:** 3
**Contribution:** 2
**Rating:** 2
**Confidence:** 5

**Summary:**

This paper proposes WSA, a theoretically motivated variant of Transformer self-attention that accounts for correlations among context vectors. The authors argue that standard attention treats context vectors as independent, an assumption that contradicts linguistic dependencies, and derive a whitening transformation based on covariance modeling. The core idea is approximating whitening via a recursive formulation (Eq. 9) that can be trained jointly with the model. Experiments with a 2-layer GPT-like model trained on the collected works of Dickens demonstrate perplexity reduction over standard self-attention, significant reduction in compute time to reach comparable validation loss, and that whitening improves decorrelation metrics (Fig. 4a) and stationarity of early-layer inputs (Fig. 4b). The authors argue this supports both theoretical and empirical efficiency improvements and could have implications for interpretability studies of trained Transformers.

**Strengths:**

- Mathematically principled derivation connecting self-attention to optimal linear estimation under correlated contexts (Secs. 3-4).

- Clear explanation of bias from correlated context vectors, intuitively shown in Figure 1.

- Empirical evidence that whitening improves convergence speed and decorrelation metrics, yet on toy scale experiments (Fig. 4a, Tabs. 1-2).

- Transparent discussion of limitations and future work (Sec. 7), including computational scalability via prefix-sum parallelization.

**Weaknesses:**

-  The experimental validation is extremely narrow, as all results are on a character-level Dickens corpus using a 1.6-1.9M parameter GPT with 2 decoder blocks. This corpus has short range dependencies and very limited vocabulary (93 tokens), which aligns well with the model’s covariance assumptions. It is unclear whether the same improvements would hold for tokenized or multilingual corpora with complex co-occurrence structures. Hence, while results are strong within this setup, the generalization to realistic language modeling conditions is questionable.

-  The whitening is derived assuming first-order stationarity and modeled with a block tridiagonal covariance. Figure 4b shows this assumption breaks down for the second decoder block, where non-stationarity increases drastically. This empirical evidence implies the whitening process fails in deeper layers, hence its impact to realistic multi-layer Transformers is again questionable.

- Even with block simplifications, recursive whitening introduces additional parameters and potential gradient instability. The authors acknowledge whitening recursion grows with sequence length and is not GPU-friendly, and suggest prefix-sum parallelization (Sec. 7), but this is prospective, not demonstrated. The method’s scalability to long contexts (e.g., N > 2048) remains speculative.

- The comparison is limited to vanilla SA with no comparison to existing orthogonalization or other-related methods. For example, orthogonal attention regularization (Xiao et al., ICML 2024).

Minor:
- Empirical results are heavily overplotted (Figure 3) without error bars or multiple runs.
- Table 1 and Figure 3 show large apparent speedups (“91% reduction”), but training time comparisons are based on different batch sizes and parameter counts (1.62 M vs 1.88 M weights, 256 vs 128 batch). Per-iteration cost is actually far higher for WSA, and the runtime savings come primarily from faster convergence under this specific loss surface. Without reporting total FLOPs or wall-clock time normalized by model size, claims of “91% reduction in training time” might be misleading.

**Questions:**

Please see the weaknesses section. Also, could whitening be applied selectively to early layers only, given non-stationarity later?

While theoretically interesting, this paper’s empirical evaluation is too limited. WSA could become an impactful line of work with broader validation, realistic datasets, proper baselines, and statistically rigorous reporting. At present, it falls short of ICLR’s standards.

---

> ### Author Response · Authors · 2025-11-20
> **Theoretical Contributions and Experimental Scope of Our Work**
>
> ## Weaknesses:
>
> The experimental validation is extremely narrow...
>
> > Yes, the experimental results are narrow, but our contribution is
> theoretical and as such, the experimental objective is validation of
> the concept, not generalization.  The extension to more complex
> corpora or tokenization schemes is a separate scientific endeavor.
> LeNet was an example of a small model of just 60,000 weights applied
> to the MNIST dataset of just 60,000 tiny (28x28) images.  It was only
> with the invention of residual connections, that the results from
> LeNet were extended to larger, more complex images and more demanding
> downstream applications.  Our view is that the goal of experiments for
> theoretical contributions is to demonstrate the derived techniques are
> valid.  This is a different scientific goal than showing a model
> generalizes for a wide swath of cases.  To this end, we have
> demonstrated that our approach leads to a theoretically justified,
> novel architectural component that significantly improves results over
> the standard GPT model in at least one case.
>
> The whitening is derived assuming first-order stationarity and modeled
> with a block tridiagonal covariance...
>
> > For this corpus, our experiments show that the whitening filter
> succeeds in decomposing the input embeddings into a first-order
> stationary process (captured by the first layer) and a non-stationary
> (or much less stationary) process.  For the moment, it is unclear
> whether stationarity will break down in more sophisticated
> architectures and more complex corpora.  It could be that multiple
> levels of whitening are required for some datasets, or that a higher
> order of whitening, say a block pentadiagonal matrix, would improve
> the response of later layers.  Regardeless of how many layers benefit
> from whitening, our paper shows that with only a single layer
> responding, the drop in loss is greatly accelerated in comparison to
> the standard model.  Exactly how this plays out over additional
> layers, more complex data, and tokenization schemes is an important
> engineering step, and is part of the next stage of our work.
>
> Even with block simplifications, recursive whitening introduces
> additional parameters and potential gradient instability...
>
> > We agree that that possibility exists.  That said, all engineering
> is about finding solutions to the types of problems you mention,
> adapting simpler solutions to more complex situations.  As noted
> above, our paper derives a method that has shown itself to be both
> stable and effective at improving performance for a small problem
> formulation, and this has significant scientific value in its own
> right.
>
> The comparison is limited to vanilla SA with no comparison to existing
> orthogonalization or other-related methods...
>
> > The paper you cite is a technique for expanding the softmax as an
> orthogonal series of terms.  Although the word orthogonalization is
> used, its application is not relevant to the theoretical insight we
> explore in our paper.  Whitening (orthogonalization) is a well-known
> and widespread technique in stochastic signal processing, and after
> an in-depth literature search, we were surprised that it has not been
> covered as a method for more efficiently processing sequences of
> embedding vectors.
>
> ## Minor
>
> Empirical results are heavily overplotted (Figure 3) without error
> bars or multiple runs...
>
> > We agree that the results compared in this paper do not compare
> flops.  Nor do they compare the amount of VRAM.  That said, wall-clock
> time is a meaningful engineering criterion.  Time, compute, memory...
> these are the three resources that affect engineering implementations.
> The premise of our theoretical work is that the whitening of the input
> sequences improves the GPT prediction of the next token.  Our
> experimental results show that the method improves the performance by
> a large margin, and at the same time the loss curve drops much more
> quickly.  These results support our insight.
>
> ## Questions:
>
> Could whitening be applied selectively to early layers only, given
> non-stationarity later?
>
> > Yes, the whitening could be applied selectively, and this is one of
> the engineering directions we intend to follow in generalizing our
> theoretical work to more general cases.
>
> While theoretically interesting, this paper’s empirical evaluation is
> too limited...
>
> > The official ICLR call for papers explicitly lists "learning
> theory" and "probabilistic methods" as two of the primary topics for
> submissions, including foundational questions about representations.
> The core focus of our paper is a foundational question about
> representation and brings to bear both theoretical and probabilistic
> contributions whose validity is confirmed by considerable experimental
> analysis.  We believe that the empirical evaluation is appropriate for
> a theoretical paper, and generalizing the theory to broader contexts
> is an important engineering objective, but not typically the
> responsibility of a theory paper.

---

> > ### Comment · Reviewer_3wfN · 2025-11-27
> > **Response**
> >
> > I thank the authors for their response. However, my main concern regarding the very narrow scope of the evaluation remains unaddressed, which continues to raise doubts about the real practical impact of WSA.
> >
> > While I understand the authors' position that their primary focus is on the theoretical side, the paper proposes a modification to the core architecture of self-attention mechanisms, one of the most critical components in modern deep learning. Given this, the evaluation feels incomplete, especially considering the limited scale of the experiments, which appear more illustrative than substantive.
> >
> > If WSA is indeed a promising direction, I believe the experimental section should be significantly strengthened to support that claim. In my view, the paper does not focus purely on theoretical analysis. Rather, it introduces a method that aims to have practical utility. This objective cannot be convincingly achieved without a more robust and comprehensive empirical validation.
> >
> > As a gesture of goodwill, I have increased my score. That said, I strongly encourage the authors to explore the practical promise of their approach more deeply, so that the community might be better positioned to assess and potentially adopt it. The field has moved beyond accepting underevaluated claims. Empirical support is not just appreciated, it is expected.
> >
> > If the authors, other reviewers, or the AC believe I’m mistaken, I won’t stand in the way. However, I cannot champion the paper in its current form.

---

### Official Review · Reviewer_xbr2 · 2025-10-31

**Soundness:** 4
**Presentation:** 2
**Contribution:** 2
**Rating:** 6
**Confidence:** 5

**Summary:**

This paper presents Whitened Self-Attention (WSA), a modification to the standard self-attention (SA) mechanism in Transformer architectures. The authors' core thesis is that standard self-attention is suboptimal as it consider context vectors as independent. The authors argue that these vectors are highly correlated, and standard attention's pairwise, weighted-sum approach (as seen in Equation 1) leads to "double-counting" of information and a biased, inefficient representation.

The authors propose to apply a whitening transform to the context vectors before they are fed into the attention mechanism. This transform decorrelates the vectors, making them stochastically orthogonal, which allows the subsequent weighted-sum operation to be an optimal linear estimator.

Recognizing that computing the full whitening matrix is computationally infeasible for large sequences, the authors derive a practical implementation.

Experiments on a small GPT model trained on a character-level dataset show promising results int terms of performance and efficiency.

**Strengths:**

Promising experimental results: The authors do a great job at motivating the community to further explore their methodology through demonstrating strong results. The performance improvements are not trivial indeed - a 91% reduction in compute time to reach an equivalent loss is great.
Pragmatic and realistic derivation: The authors manage to nicely bridge the gap between the theoretically motivated whitening (but computationally impossible) solution and a practical, implementable one. The step-by-step simplification from the full covariance matrix is logical and well-explained.
Interesting analysis of the whitening process: It is appreciated that the authors did not only present results from the Whitening filter, but also empirically validated how the whitening impacts outputs through training.

**Weaknesses:**

Scale of experiments very limited: While I full agree with the statement of the authors in the discussion regarding the importance of having controlled experiments in smaller setting, this particular setup is far from representative of the conditions under which GPT architectures demonstrate their characteristic behavior. The use of a ~1 M parameter model and a single corpus (Dickens) makes it impossible to infer how the proposed whitening mechanism would behave at scale. Given that Transformer performance, optimization dynamics, and attention patterns fundamentally change with model size, the presented results should be interpreted as conceptual evidence rather than empirical validation of scalability.
The writing would benefit from better structuring: I found the result section particularly convoluted to parse, with numerous results presented in succession, spread accross multiple tables. Furthermore, methodology of the experiments (e.g. batch size, number of parameters) was also merged with results and analysis - sometimes within the same paragraph. I recommend the use of subsections and perhaps focusing on the most important results and leaving the rest for a supplementary material.

**Questions:**

I am particularly confused by the choice of character-level tokenization - particularly in the context of the paper which aims to whiten (i.e. decorrelating) the input vectors. Character-level sequences inherently exhibit lower semantic correlation than word- or subword-level embeddings, which makes them a poor test bed for studying correlation structures in attention. Wouldn’t the results be more meaningful if demonstrated with more standard tokenization or embedding strategies, where inter-token correlations play a much greater role? Since it is such an uncommon choice (and combined with the already small scale of the experiments) I would expect either additional experiments with standard tokenization or a stronger justification for why character-level data is appropriate for evaluating the proposed method.

---

> ### Author Response · Authors · 2025-11-20
> **Scaling Considerations and Tokenization Choices Explained**
>
> ## Weaknesses:
>
> Scale of experiments very limited: While I full agree with the
> statement of the authors in the discussion regarding the importance of
> having controlled experiments in smaller setting, this particular
> setup is far from representative of the conditions under which GPT
> architectures demonstrate their characteristic behavior. The use of a
> ~1 M parameter model and a single corpus (Dickens) makes it impossible
> to infer how the proposed whitening mechanism would behave at
> scale. Given that Transformer performance, optimization dynamics, and
> attention patterns fundamentally change with model size, the presented
> results should be interpreted as conceptual evidence rather than
> empirical validation of scalability. The writing would benefit from
> better structuring: I found the result section particularly convoluted
> to parse, with numerous results presented in succession, spread
> accross multiple tables. Furthermore, methodology of the experiments
> (e.g. batch size, number of parameters) was also merged with results
> and analysis - sometimes within the same paragraph. I recommend the
> use of subsections and perhaps focusing on the most important results
> and leaving the rest for a supplementary material.
>
> > We agree that the scale of the experiments is small.  That said, our
> architecture was designed in accordance to the Chinchilla scaling
> rules of thumb. These state that there should be roughly 20 tokens per
> model parameter.  This implies our model should have on the order of
> 1m weights split about equally between the embeddings and the
> remaining model weights.  Under these design specifications, two
> layers with two attention heads per layer makes sense.  Although we
> did not provide an example from the model used in inference, we have
> found that the trained model using this simple character tokenization
> scheme produces syntactically convincing output, even capturing
> stylistic characteristics of Dicken's writing.  If you think it useful
> we could add sample output to an appendix.
>
> > We also agree that our work
> does not demonstrate scalability.  Rather, it is primarily a
> theoretical contribution, conceptually validated with experiments.
> Small models with limited results have, historically, been impactful.
> For example, LeNet had only 60,000 weights applied to a dataset of
> 60,000 28x28 images (MNIST).  It was only with the later invention of
> residual connections that the results from LeNet were
> extended to larger datasets, more complex images, and more
> sophisticated downstream applications.
>
> > Many thanks for your feedback on the results section.  We will
> improve the presentation in that section and upload an improved draft.
>
> ## Questions:
>
> I am particularly confused by the choice of character-level
> tokenization - particularly in the context of the paper which aims to
> whiten (i.e. decorrelating) the input vectors. Character-level
> sequences inherently exhibit lower semantic correlation than word- or
> subword-level embeddings, which makes them a poor test bed for
> studying correlation structures in attention. Wouldn’t the results be
> more meaningful if demonstrated with more standard tokenization or
> embedding strategies, where inter-token correlations play a much
> greater role? Since it is such an uncommon choice (and combined with
> the already small scale of the experiments) I would expect either
> additional experiments with standard tokenization or a stronger
> justification for why character-level data is appropriate for
> evaluating the proposed method.
>
> > We agree that characters alone carry less semantic meaning than
> words.  That said, semantic relations are not the only correlations
> found in written language, as letters are a foundation of syntax.
> Furthermore, character correlations have been successfully
> exploited by file compression applications.  A great many papers have
> been written on the application of neural networks to bits per character compression, and have successfully achieved SOTA levels of of file compression.

---

### Official Review · Reviewer_k8zr · 2025-11-04

**Soundness:** 2
**Presentation:** 2
**Contribution:** 2
**Rating:** 2
**Confidence:** 4

**Summary:**

The paper is about improving attention in transformer models. The main observation is that token embeddings are temporally correlated, and the authors posit that removing these correlations would improve training efficiency. They achieve this by applying a temporal "whitening" transformation right before each attention layer.

The authors train a very small transformer model to evaluate their approach. The results show a dramatic improvement in validation perplexity. However, more experimental work to establish confidence in the method is required before it is published at a top tier conference.

**Strengths:**

* The experimental results show very strong perplexity improvements.

* The exposition is clear, and a good discussion of extensions and limitations in Sec 7. The authors' perspective of whitening is interesting.

**Weaknesses:**

* The transformer model is too simplistic. I believe the model size of 2m params is too small to give confidence in the results. In addition, there is two model properties that might confound the results: 1. baseline model only has two layers, while the proposed model is very sequential since tokens are processed sequentially, possibly making its effective depth larger. 2. the single-character tokenizer is too weak, and there is a possibility that the proposed technique gives strong improvement because of that (this is actually suggested by the authors' results that the whitening is strong in layer 1 but not layer 2). The authors should evaluate using deeper models and real tokenizers.

* I did not find the theoretical or empirical motivation of why tokens need to be whitened. It would help to have some experiments to prove that this is an issue. Also the authors make multiple logical leaps on the covariance structure (e.g. tridiagonal structure) that are not backed by data.

* I did not see any ablations.

**Questions:**

Q: Did the authors tune the learning rate of the baseline?

---

> ### Author Response · Authors · 2025-11-20
> **On the Role of Theoretical Contributions and their Experimental Scope**
>
> ## Weaknesses
>
> The transformer model is too simplistic. I believe the model size of
> 2m params is too small to give confidence in the results.
>
> > The main contribution of our paper is theoretical.  Our view is that
> theory papers should demonstrate validity of new concepts, not
> generalize methods.  Historically, neural models have gone through
> many iterations to make them more generally applicable.  An example,
> LeNet, was a small model of just 60,000 weights that worked on a small
> dataset of 60,000 28x28 images (MNIST).  Residual connections,
> invented later, were needed to extend LeNet to more complex problems.
> We expect that our technique will likely require additional work to
> achieve its full potential, but that its theoretical insights and
> engineering solutions are significant contributions.  With regards to
> the model size, we based our design on the Chinchilla scaling rule of
> thumb, which state that there should be roughly 20 tokens per model
> weight.  Given the corpus has 13.8m characters, a model size on the
> order of 0.7m tokens should suffice, and ours was roughly double that.
> We believed that our approach was thoughtful and pragmatic but would
> welcome any additional guidelines you could provide on how to
> determine adequate model sizes.
>
> In addition, there is two model properties that might confound the results:
> 1. baseline model only has two layers, while the proposed model is
> very sequential since tokens are processed sequentially, possibly
> making its effective depth larger.
>
> 2. the single-character tokenizer is too weak, and there is a
> possibility that the proposed technique gives strong improvement
> because of that (this is actually suggested by the authors' results
> that the whitening is strong in layer 1 but not layer 2). The authors
> should evaluate using deeper models and real tokenizers.
>
> > Your intuition about tokenization may be justified for some
> applications, but there are many application areas benefiting from character
> tokenizations strategies.  An example is file compression.  For
> example, character tokenization was used by Al-Rfou et al. in
> **Character-Level Language Modeling with Deeper Self-Attention**.  We
> have provided both a rationale and experimental evidence that explain
> the behavior of the whitening filter in the first layer vis-a-vis the
> second one.
>
> I did not find the theoretical or empirical motivation of why tokens
> need to be whitened. It would help to have some experiments to prove
> that this is an issue. Also the authors make multiple logical leaps on
> the covariance structure (e.g. tridiagonal structure) that are not
> backed by data.
>
> > Notionally, the structure found in sequential data can be decomposed
> into stationary and nonstationary parts.  The stationary part
> corresponds to what can be modeled as a stochastically driven linear,
> time-invariant system.  It is possible that some sequences do not have
> any stationary component, and in this case, our approach would provide
> no value.  However, at the character tokenization level, our paper
> shows that there is a stationary component that is well represented and
> successfully separated out by the first layer of the model.  As shown
> in Figure 4a, the output of the first whitening filter is indeed
> significantly whiter than the input.  This in conjunction with the
> accelerated decrease in validation loss is evidence that the theory
> and engineering approximations are valid.Furthermore, Figure 4b shows
> that the first-order stationarity of the first layer's input is is
> measurably stronger than that for the second layer, supporting our
> view that input can be decomposed into stationary and non-stationary
> parts.
>
> > With regards to the to the theoretical motivation for whitening of
> tokens, Section 2 of our paper develops both the intuition and
> benefits expected from whitening, and our experimental results
> illustrate that the implementation succeeds in reducing the loss much
> faster (in both iterations and wall-clock time) with respect to the
> standard model. All the engineering approximations we employ are drawn
> from the literature on stochastic signal processing.  Our sources are
> cited in the paper.  Let us know if you think the referencing is
> insufficient or whether the background material should be recapped in
> an appendix.
>
> I did not see any ablations.
>
> > Our paper studies the simplest possible model (two layers, two
> attention heads per layer) with the addition of one single,
> theoretically motivated component, the whitening filter. We compare
> this to the model without that component.  As ablations are for
> determining which components can be removed without adverse effect, it
> is not clear to us why it is needed for this case, but would
> appreciate your insights on this question.
>
> ## Questions
>
> Q: Did the authors tune the learning rate of the baseline?
>
> > No, we used a learning rate that is commonly used for character
> tokenization, and we used the same learning rate for all the cases.

---

### Author Response · Authors · 2025-11-30
**The Role of Theory in Deep Learning**

Currently, deep learning is a field strongly oriented toward experimental work, often undervaluing theory. The community rewards practical, generalizable results that extend prior empirical findings. Yet mature science depends equally on theoretical models, which deepen understanding, refine intuition, and support prediction and analysis. Theory and experiment together form a virtuous cycle: experiments generate observations, theory explains them, further experiments probe the limits of those explanations, and the results in turn sharpen the models.

Deep learning architectures in widespread use are widely acknowledged to be poorly understood, raising concerns about safety, reliability, and reproducibility. Theoretical modeling is essential for addressing these concerns, but realizing its benefits requires close collaboration between experimental and theoretical work. This paper contributes a theoretical insight about self-attention, supported by experiments that validate its core predictions. Reviewers have recognized that the work is theoretically sound and well written, but have questioned the scale of the model and the breadth of the results. We contend that, despite these concerns, the paper makes a substantial and timely theoretical contribution to deep learning, and that such work is crucial in a field where theory remains underrepresented.

---

### Meta-Review · Area_Chair_sEbi · 2026-01-13

**Summary:**

Reviewers agree the idea is interesting and the reported gains on the authors’ controlled setup are large, but they consistently question whether the paper meets ICLR’s bar given the narrow empirical scope, missing baselines/ablations, and several methodological/theoretical gaps that affect confidence in the claims.

- **Scope and representativeness of experiments**: Multiple reviewers emphasize that results are shown only on a very small 2-layer, ~1–2M parameter character-level GPT trained on Dickens, which is “too small to give confidence” (Reviewer k8zr) and “far from representative” such that it is “impossible to infer how the proposed whitening mechanism would behave at scale” (Reviewer xbr2). Reviewer 3wfN similarly calls the validation “extremely narrow” and argues generalization to “realistic language modeling conditions is questionable” (Reviewer 3wfN).

- **Missing ablations and insufficient baselines**: Reviewer k8zr states “I did not see any ablations” (Reviewer k8zr). Reviewer 3wfN notes the comparison is limited to vanilla self-attention “with no comparison to existing orthogonalization or other-related methods” (Reviewer 3wfN) and flags reporting issues (e.g., “overplotted … without error bars,” and that training-time claims may be “misleading” due to different batch sizes/parameter counts) (Reviewer 3wfN).

- **Theory/assumption validation and confounds**: Reviewers challenge whether key assumptions and design choices are adequately justified or validated. Reviewer k8zr reports “multiple logical leaps on the covariance structure (e.g. tridiagonal structure) that are not backed by data” and lacks “theoretical or empirical motivation” (Reviewer k8zr). Reviewer g2W4 questions the “assume” steady-state replacement and asks how whitening is encouraged “without any regularization loss” (Reviewer g2W4), and later questions whether Eq. 9 is necessary versus “simply adding a layer of RNN before standard attention” (Reviewer g2W4). Reviewers also note potential confounds (effective depth / tokenization) and request deeper models and standard tokenizers (Reviewer k8zr; Reviewer xbr2; Reviewer 3wfN).

Taken together, the reviews suggest the work is a promising direction but currently under-validated and under-controlled to support strong claims of efficiency/scalability/generalization for a core architectural change, motivating a reject recommendation at this stage.

**Reviewer Concerns:**

## Concerns addressed (partially) by the rebuttal/discussion:

- **Clarification of the “steady-state” approximation and its provenance**: In response to concerns about “assum[ing] … converges to a steady state” (Reviewer g2W4), the authors clarify “we are not assuming, we are approximating,” citing ARMA/LTI stability intuition and noting the steady-state relates to an algebraic Riccati equation (Authors’ reply to Reviewer g2W4). This improves interpretability of the derivation, though it remains largely an argument from established control/time-series practice rather than evidence specific to the proposed setting.

- **Why whitening emerges without an auxiliary whitening loss**: The authors state they “expected it would be necessary to construct an auxiliary loss … but … discovered that they converge without it,” and point to Fig. 4a/4b as evidence (Authors’ reply to Reviewer g2W4). This directly answers the “how do you encourage them to converge” question (Reviewer g2W4), but does not address comparative necessity versus simpler alternatives.

- **Tokenization choice and small-scale justification**: Reviewers were “confused by … character-level tokenization” (Reviewer xbr2) and worried it is “too weak” and may confound results (Reviewer k8zr). The authors justify character-level modeling as relevant for syntax/compression and cite precedent (e.g., character-level language modeling work) (Authors’ replies to Reviewers k8zr and xbr2). They also justify model size via a Chinchilla-style token/parameter heuristic (Authors’ replies to Reviewers k8zr and xbr2). This explains rationale but does not remove the reviewers’ core request for evidence on standard tokenization and larger/deeper settings.

- **Scalability framing**: Several reviewers ask for scale evidence; authors explicitly concede “our work does not demonstrate scalability” and frame the experiments as conceptual validation for a theoretical contribution (Authors’ replies to Reviewers xbr2 and 3wfN). This addresses expectation-setting but not the underlying empirical concern.

## Concerns still outstanding (and central to rejection):

- **Empirical breadth and robustness remain insufficient**: The primary criticism—evaluation is “extremely narrow” (Reviewer 3wfN) and “too small to give confidence” (Reviewer k8zr)—was not resolved with new experiments. Reviewer 3wfN reiterates post-rebuttal that the “very narrow scope … remains unaddressed,” calling the evaluation “incomplete” for a method intended to have practical utility (Reviewer 3wfN, post-rebuttal comment).

- **Missing baselines against plausible alternatives**: Reviewer g2W4 explicitly asks why Eq. 9 is preferable to “a layer of RNN … LSTM/GRU … Linear … CNN before the standard attention” (Reviewer g2W4). The authors respond it is “conceivable — even likely that an RNN could deliver similar results” (Authors’ reply to Reviewer g2W4), but no comparative experiments/ablations are provided. Similarly, Reviewer 3wfN requests comparisons to “existing orthogonalization or other-related methods” (Reviewer 3wfN), which are not supplied.

- **Incomplete ablations and confound controls**: Reviewer k8zr states “I did not see any ablations” (Reviewer k8zr) and raises confounds (effective depth; tokenizer) (Reviewer k8zr). The authors argue ablations are unclear when only one component is added (Authors’ reply to Reviewer k8zr), but this does not satisfy the community-standard expectation to control alternative explanations (e.g., sequential processing depth, parameter/batch-size differences, and learning-rate tuning). Reviewer 3wfN also flags that time/efficiency claims may be “misleading” due to differing batch sizes/parameter counts and asks for FLOPs/normalized reporting (Reviewer 3wfN), which remains unresolved.

- **Validation of modeling assumptions with data**: Reviewer k8zr highlights “logical leaps on the covariance structure … not backed by data” (Reviewer k8zr). Reviewer 3wfN also notes stationarity assumptions degrade in deeper layers (“assumption breaks down for the second decoder block”) and questions applicability to realistic multi-layer transformers (Reviewer 3wfN). The rebuttal largely frames this as future engineering (e.g., higher-order whitening; selective early-layer whitening), rather than providing evidence that assumptions hold broadly (Authors’ reply to Reviewer 3wfN).

Overall, the rebuttal improves clarity and acknowledges limitations, but the key acceptance blockers—limited scope, missing baselines/ablations, and insufficient evidence supporting generality and necessity of the proposed mechanism—remain outstanding.

**Reviewer Scores:**

- **Reviewer k8zr (initial Rating 2, Confidence 4): Likely unchanged (2)**. Their main requests—testing “deeper models and real tokenizers,” providing motivation/validation for covariance assumptions (“logical leaps … not backed by data”), and adding ablations—were not met with additional experiments (Reviewer k8zr; Authors’ replies primarily provide rationale rather than new evidence).

- **Reviewer xbr2 (initial Rating 6, Confidence 5): Likely unchanged (6)**. This reviewer already characterizes the work as promising but states the scale “makes it impossible to infer … at scale” and flags character tokenization as an atypical/weak testbed (Reviewer xbr2). The authors acknowledge they “did not provide” scale evidence and position the results as conceptual validation (Authors’ reply to Reviewer xbr2). This likely does not move the reviewer materially, and they already note they “would not mind if paper is rejected” (Reviewer xbr2).

- **Reviewer 3wfN (initial Rating 2, Confidence 5): Would increase modestly but remain below/at rejection (estimated 4)**. The reviewer explicitly states: “As a gesture of goodwill, I have increased my score,” yet reiterates that the “very narrow scope of the evaluation remains unaddressed” and that the paper “cannot be convincingly achieved without a more robust and comprehensive empirical validation,” concluding “I cannot champion the paper in its current form” (Reviewer 3wfN, post-rebuttal comment). This points to a small upward adjustment but not to an accept-level endorsement.

- **Reviewer g2W4 (initial Rating 4, Confidence 4): Likely unchanged (4)**. After author clarification, the reviewer’s remaining concern is the necessity of Eq. 9 versus simpler learned modules (“why … over … a one-layer LSTM/GRU … Linear … CNN”) (Reviewer g2W4). The authors concede this is “conceivable — even likely” (Authors’ reply to Reviewer g2W4) and do not provide comparative baselines, so a score increase is unlikely.

---

### Decision · Program_Chairs · 2026-01-26

Reject